# REPRESENTATION-SPACE DIFFUSION MODELS FOR GENERATING PERIODIC MATERIALS

## ABSTRACT

Generative models hold the promise of significantly expediting the materials design process when compared to traditional human-guided or rule-based methodologies. However, effectively generating high-quality periodic structures of materials on limited but diverse datasets remains an ongoing challenge. Here we propose a novel approach for periodic structure generation which fully respect the intrinsic symmetries, periodicity, and invariances of the structure space. Namely, we utilize differentiable, physics-based, structural descriptors which can describe periodic systems and satisfy the necessary invariances, in conjunction with a denoising diffusion model which generates new materials within this descriptor or representation space. Reconstruction is then performed on these representations using gradient-based optimization to recover the corresponding Cartesian positions of the crystal structure. This approach differs significantly from current methods by generating materials in the representation space, rather than in the Cartesian space, which is made possible using an efficient reconstruction algorithm. Consequently, known issues with respecting periodic boundaries and translational and rotational invariances during generation can be avoided, and the model training process can be greatly simplified. We show this approach is able to provide competitive performance on established benchmarks compared to current state-of-the-art methods.

## 1 INTRODUCTION

Discovering or designing new materials with potentially useful functional properties remains a long-standing challenge in the material sciences. Traditionally, this is accomplished by the exhaustive experimental or computational evaluation of materials within a synthetically amenable chemical space (Ceder, 1998; Alberi et al., 2018; de Pablo et al., 2019; Reymond, 2015). However, this process is highly time-consuming and costly, and requires covering a nearly infinite chemical space, therefore there is a strong demand for quicker and more efficient alternatives. While recent developments in generative models have demonstrated great promise in other domains such as towards protein and molecular design, similar advancements in materials generation remain limited and under-explored. This stems from the requirement that it is not sufficient for the generative model to only provide the chemical composition of the material, but it should also generate the exact three-dimensional atomic structure defined by a set of Cartesian coordinates within a unit cell (Pauling, 1929; Butler et al., 2016), which plays a key role in determining the material's properties (Oganov et al., 2019; Price, 2014). In addition, generative models for materials should also be robust to data scarcity, as materials datasets typically contain up to tens or hundreds of thousands of training samples, far fewer than the millions and billions of samples in other domains such as computer vision and natural language processing. These limitations have led to the development of new approaches that can address the symmetries, periodicity, and invariances inherent in periodic structures, and which perform significantly better than methods which can not, with rare exceptions.

Generative models for periodic materials can be broadly categorized into two groups, representation-based and direct generation. In the former, materials are first encoded into suitable representations, which then serve as the training dataset for a generative model. Subsequently, new representations generated by the trained model are decoded back to the Cartesian space within a unit cell. Such an approach is analogous to the generation of molecular graphs(Simonovsky & Komodakis, 2018; De Cao & Kipf, 2018) or SMILES strings(Gómez-Bombarelli et al., 2018; Kotsias et al., 2020) for molecules. However, since molecular graphs or SMILES lack spatial information, other represen-

tations are needed, which remains an ongoing challenge for this method. In comparison, the latter approach directly generates atomic coordinates in the Cartesian space, either in a one-shot manner or through an iterative process such as denoising. In this approach, the model and the generative process would need to respect periodicity and invariances of the structure, which is not always satisfied in current works.

In this study, we present **Struct**ure **Rep**resentation **Diff**usion (StructRepDiff), a novel approach for materials generation that not only overcomes the constraints of existing representation-based methods, but also leverages on the generative performance of denoising diffusion models commonly seen in direct generation. Specifically, we employ differentiable, physics-based structural descriptors as representations and perform denoising in the representation space. As these descriptors are constructed to inherently respect periodicity and invariance, there is no requirement for special considerations such as a custom score function, thereby greatly simplifying the training process. We concatenate the descriptor with composition and cell vectors to create our final representation vector. To reconstruct or generate three-dimensional structures, we then utilize an efficient reconstruction algorithm that builds three-dimensional structures from the representation space. We propose the novel architecture to implement our Gradient-based structure reconstruction from representation space to structure space. To our best knowledge, this is the first example of periodic materials generation with diffusion models in the representation space rather than the Cartesian space. Fig. 1 provides an overview of StructRepDiff.

**Main Contributions.** In this paper, we present our approach StructRepDiff with the following main contributions:

- We propose a novel approach to material generation in the representation space, utilizing a denoising diffusion model commonly employed in direct generation.
- We propose a working invariant representation for materials capable of generating structures in the representation space, demonstrating competitive performance relative to existing methods. As discussed in details in section 4.1.
- We propose a novel implementation of a gradient-based reconstruction method for material reconstruction from our proposed representations. As discussed in details in section 4.3.

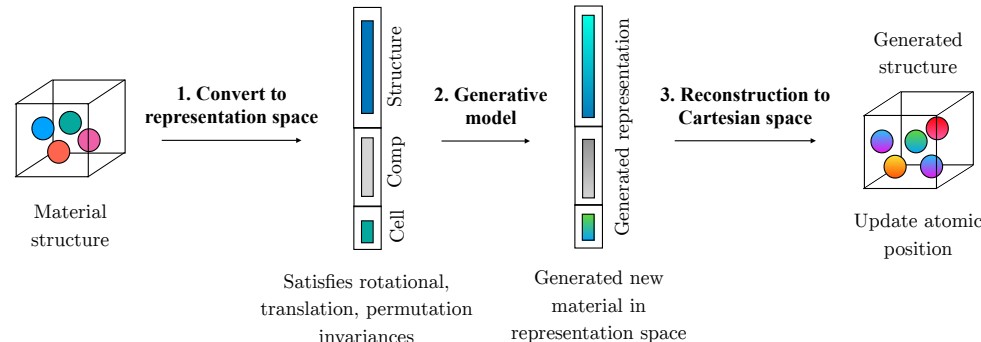

Figure 1: An overview of the StructRepDiff process. The vectorial structure representation $R$ from section 4.1 is schematically shown in captions as Cell, Comp and Structure for $L$, $C_{\text{comp}}$ and $R_{\text{str}}$ respectively.

## 2 RELATED WORKS

### 2.1 REPRESENTATION-BASED MATERIALS GENERATION

In representation-based materials generation, a suitable encoding or representation of materials is critical for an effective generation. Specifically, the suitability of a representation is determined by

whether this representation (1) can describe uniquely the Cartesian positions, unit cell, and composition of a given material, (2) is invariant to translations, rotations and permutations, and most importantly, (3) is invertible back to the Cartesian space. Musil et al. (2021); Noh et al. (2020) It is important to note that presently, there is no single existing representation that fulfills all three of the conditions mentioned above, despite numerous ongoing efforts to develop one. Consequently, in all current cases of representation-based generation, there will be shortcomings in meeting one or more of these criteria, which can impede the effectiveness of this approach. For instance, voxel-based representations, as proposed by Hoffmann et al. (2019) and Noh et al. (2019), lack both translational and rotational invariance. These representations are also susceptible to finite spatial resolution issues and struggle to efficiently capture the compositional dimension. The work presented by Uhrin (2021) in exploring the structure of representation space is quite noteworthy. While the study did not integrate these representations into a generative model, which somewhat limits their immediate applicability, the insights gained are valuable.

Our approach differs from previous methods here by enforcing a strict rule for our representation to follow conditions (1) and (2), even if it does not intrinsically meet the invertibility requirement in (3). Instead, we use a separate reconstruction process to satisfy invertibility, which we will describe in the Methods section.

## 2.2 DIRECT MATERIALS GENERATION

In the case of direct generation, a generative model decodes the Cartesian positions, unit cell, and composition directly through various means. Most notable are a class of approaches which make use of diffusion models to denoise a set of noisy positions to positions resembling a real material. This was first demonstrated by Xie et al. (2021) in the CDVAE model. In this approach, a graph neural network (GNN) is used to encode the material into a graph-level embedding, which is separately used to predict the number of atoms and composition of the material, and used as the condition in a noise-conditional score network to denoise the positions and atomic numbers. A subsequent approach by Luo et al. (2023) builds on this method by introducing a modified approach for predicting atomic numbers and a modified score function for incorporating translational invariance in the denoising process. Different from these approaches, we perform denoising on the representation instead of the Cartesian positions, as well as on the composition and lattice parameters directly.

## 3 BACKGROUND

An arbitrary material structure $M$ with $n$ atoms can be fully described by using three components: 1) atomic numbers $\boldsymbol{A} = (a_1, \ldots, a_n)$, where $a_i \in \mathbb{A}$ and $\mathbb{A}$ is the set of all chemical elements; 2) atomic positions $\boldsymbol{X} = (\boldsymbol{x}_1, \ldots, \boldsymbol{x}_n)$, where $\boldsymbol{x}_i \in \mathbb{R}^3$ is the Cartesian coordinates of atom $a_i$; and 3) lattice parameters $\boldsymbol{L} = (l_1, l_2, l_3, \gamma_1, \gamma_2, \gamma_3)$, where $l_1$, $l_2$ and $l_3$ are the unit cell lengths and $\gamma_1$, $\gamma_2$ and $\gamma_3$ are the angles between the lattice vectors of the unit cell. In the context of generating a periodic material structure $M = (\boldsymbol{A}, \boldsymbol{X}, \boldsymbol{L})$, it is necessary to generate all three components.

### 3.1 MATERIAL STRUCTURE REPRESENTATION

While the atomic positions $\boldsymbol{X}$ specifies the material structure, it is generally an undesirable choice of encoding or representation to be directly employed in data-driven approaches for materials generation. Specifically, this is due to their lack of translation, rotation and permutation invariances. To address this limitation, we propose to use existing structural descriptors, which are mathematical representations used to describe the arrangement and electronic environments of atoms in molecules and crystal structures. While there are many descriptors available, an effective structural descriptor for materials generation should conform to translation, rotation, and permutation invariances. [Propositions 1, Proof in appendix A.1.1].

**Embedded Atom Density (EAD)**. It is worth noting that StructRepDiff is descriptor-agnostic, as long as the descriptor exhibits invariance. In this work, we have selected the embedded atom density (EAD) descriptor as an illustrative example, for it observes all three invariances required. Introduced by Zhang et al. (2019), the core concept of the EAD descriptor revolves around the notion that each atom within a system interacts with a collective electron cloud formed by its neighboring atoms. Readers interested in a more in-depth understanding of the theory can find a brief overview in Ap-

pendix A.2. We hypothesize that the invariant descriptors like EAD form representation spaces that are more effective than those from simple distances and angles which lack permutation invariance, thus leading to improved performance of generative models trained within this space.

## 3.2 GENERATIVE DIFFUSION MODEL

A generative diffusion model is implemented to learn the training structural distributions and create novel structures resembling the structures from the initial distribution. Diffusion models have shown impressive results for images Dhariwal & Nichol (2021), point clouds Cai et al. (2020); Luo & Hu (2021), and molecular structures Shi et al. (2021); Xie et al. (2021). Different variants of diffusion models exist, encompassing diffusion probabilistic models Sohl-Dickstein et al. (2015), noise conditioned score networks (NCSN) Song & Ermon (2019), and denoising diffusion probabilistic models (DDPM) (Ho et al. (2020) Ho et al., 2020). In our approach, we implement the denoising diffusion probabilistic models (Ho et al., 2020) with a goal to denoise the representation of the input materials. Starting with the input vector $x_0$, we progressively add Gaussian noise over $T$ steps.

**Forward process.** Given data-point $\boldsymbol{r}_0$ drawn from the actual data distribution $q(\boldsymbol{r})$. The updated latent variable for the next diffusion step $\boldsymbol{r}_t$ follows a distribution $q(\boldsymbol{r}_t|\boldsymbol{r}_{t-1}) = \mathcal{N}(\boldsymbol{r}_t; \mu_t = \sqrt{1-\beta_t}\boldsymbol{r}_{t-1}, \Sigma_t = \beta_t\mathbf{I})$ from appendix A.3.3.

**Reverse Process.** From the forward process, as $t \to \mathrm{T}$ approaches $\infty$, the latent $\boldsymbol{r}_T$ is nearly an isotropic Gaussian distribution. The reverse diffusion process attempts to approximate $q(\boldsymbol{r}_{t-1}|\boldsymbol{r}_t)$ with a neural network $p_\theta$ as shown in appendix A.3.3. As the reverse process is also sets up as a Markov process from $r_t \to r_{t-1} \to \dots \to r_0$. With $p_\theta(\boldsymbol{r}_T) = \mathcal{N}(\boldsymbol{r}_t, 0, I)$ being the pure noise distribution. The aim of the model being, learning $p_\theta(\boldsymbol{r}_0) = \int p_\theta(\boldsymbol{r}_{0:T})d\boldsymbol{r}_{0:T} = \int p_\theta(\boldsymbol{r}_T) \prod_{t=1}^{T} p_\theta(\boldsymbol{r}_{t-1}|\boldsymbol{r}_t)d\boldsymbol{r}_{0:T}$.

**Training.** The objective of the model is to maximise the likelihood of the generated sample $\boldsymbol{r}_0$ to be from the initial data distribution $q(\boldsymbol{r}_0)$. Which in the reverse-process framework would mean maximise the probability $p_\theta(\boldsymbol{r}_0)$, which is the marginalised probability $\int p_\theta(\boldsymbol{r}_{0:T})d\boldsymbol{r}_{0:T}$ as shown in appendix A.3.3; While learning the parameters $\theta$ of the neural network model.

## 4 METHODOLOGY

Our proposed approach, StructRepDiff, generates three-dimensional structures by first generating structure representations in the representation space via a diffusion model. Subsequently, these representations are reconstructed back to the Cartesian space through an iterative, procedure.

### 4.1 STRUCTURES TO REPRESENTATIONS

Given a material structure $M = (\boldsymbol{A}, \boldsymbol{X}, \boldsymbol{L})$, we map it to a vectorial structure representation $\boldsymbol{R} = [\boldsymbol{R}_{\mathrm{str}} \oplus \boldsymbol{C}_{\mathrm{comp}} \oplus \boldsymbol{L}] \in \mathbb{R}^N$, where $\boldsymbol{R}_{\mathrm{str}}$ is the expanded EAD representation, $\boldsymbol{C}_{\mathrm{comp}}$ is the composition representation, $\boldsymbol{L}$ is the lattice parameters or representation and $\oplus$ denotes vector concatenation.

**Expanded EAD representation.** The expanded EAD representation $\boldsymbol{R}_{\mathrm{str}}$ forms the backbone of the full structure representation $\boldsymbol{R}$. First, note that the EAD descriptor creates individual, atom-level representations $\{\boldsymbol{e}_i\}_{i=1}^n$ for a system of $|\boldsymbol{A}| = n$ atoms, where $\boldsymbol{e}_i \in \mathbb{R}^{N_e}$ with $N_e$ being the hyperparameter for the EAD representations. The total number of atoms $n$ in the material is obtained explicitly. To obtain the structure-level EAD representation $\boldsymbol{R}_{\mathrm{EAD}}$, we take max- and min-pooling over all atomic representations, before concatenating them together. Mathematically, we have

$$\boldsymbol{R}_{\mathrm{EAD}} = [\mathcal{P}_{\max}(\{\boldsymbol{e}_i\}_{i=1}^n) \oplus \mathcal{P}_{\min}(\{\boldsymbol{e}_i\}_{i=1}^n)] \in \mathbb{R}^{2N_e}, \tag{1}$$

where $\mathcal{P}$ is a pooling operation over a set of vectors. Furthermore, we also include an additional dimension to account for the minimum inter-atomic distance $d_{\min}$ present in $M$. Thus, the expanded EAD representation $\boldsymbol{R}_{\mathrm{str}} = [\boldsymbol{R}_{\mathrm{EAD}} \oplus d_{\min}] \in \mathbb{R}^{2N_e+1}$.

**Composition representation.** The composition representation $\hat{\boldsymbol{C}}_{\mathrm{comp}}$ is a normalized vector that encodes compositional information of $M$. Concretely, $\hat{\boldsymbol{C}}_{\mathrm{comp}}$ is an instance of frequency encoding of

all the atoms in $M$, normalized by the total number of atoms $n$. Arrangement of elements in $\hat{C}_{\text{comp}}$ corresponds to the periodic table, e.g. normalized count of Hydrogen is the first element in this vector. To recover the information of the material from there normalised composition, we concatenate the number of atoms $n$ with the $\hat{C}_{\text{comp}}$ vector. Thus, the expanded composition representation $C_{\text{comp}} = [\hat{C}_{\text{comp}} \oplus n] \in \mathbb{R}^{|\mathbb{A}|+1}$. In practice, we set $|\mathbb{A}| = 96$, the number of unique elements in our training dataset.

**Lattice representation.** The last piece of information needed to fully describe a material structure $M$ is the lattice parameters $L \in \mathbb{R}^6$. Here, we use $L$ directly as the lattice representation.

Concatenated together, the three representations outlined above form the full structure representation $R \in \mathbb{R}^N$, where $N = 2N_e + |\mathbb{A}| + 8$.

## 4.2 GENERATING NEW STRUCTURES IN REPRESENTATION SPACE

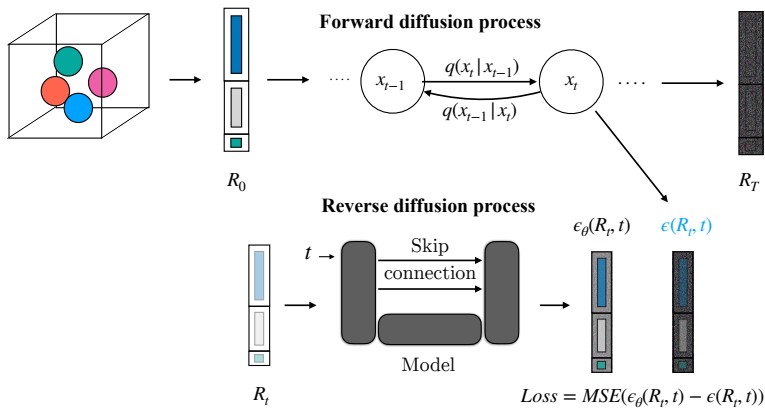

Figure 2: Training pipeline of StructRepDiff model showing the forward and reverse diffusion process. The forward diffusion process add noise to the initial representation $R_M$ in $T$ time-steps to reach to $R_T$ and the reverse diffusion model learns to remove noise by training against the actual noise $\epsilon(R_t, t)$ from the forward model. The U-net based model tries to predict $\epsilon_\theta(R_t, t)$ at every time-step $t$ during the training.

In the generative process, we aim to first generate representations of new materials before reconstructing structures from the representation space. Our representation generation can be formulated as follows: let the dataset $\mathcal{D} = \{R_0^{(j)}\}_{j=1}^m$ be the set of representations obtained from a set of $m$ material structure $\{M^{(j)}\}_{j=1}^m$. The subscript in $R_0$ indicates that this representation is at the first time step of the forward diffusion process. Our goal is to estimate the true data distribution $q(R_0)$ with our model and generate some representation $\tilde{R}_0$, where $\tilde{R}_0 \notin \mathcal{D}$. After the diffusion model has been trained according to the process in Fig. 2, we can generate new representations by successively removing noises from an initialized representation $R_T$, sampled from Gaussian noise. Here, $T$ indicates the final time step of the forward diffusion process. This process is outlined in Algo. 3 and Fig. 3. The experimental details and hyper-parameters of the diffusion model and training are discussed in appendix A.3.4.

---

**Algorithm 1** Generative diffusion sampling

---

1: $R_T \sim \mathcal{N}(0, I) \rightarrow R_T \in \mathbb{R}^N$
2: **for** $t = T, ....1$ **do**
3:     $z \sim \mathcal{N}(0, I)$ if $t > 0$, Else $z = 0$
4:     $R_{t-1} = \frac{1}{\sqrt{\alpha_t}} \left( R_t - \frac{\beta_t}{\sqrt{1 - \prod_{s=0}^t \alpha_s}} \epsilon_\theta(R_t, t) \right) + \sqrt{\beta_t} z$
5: **end for**
6: **return** $R_T \rightarrow \tilde{R}_0 \in \mathbb{R}^N$

---

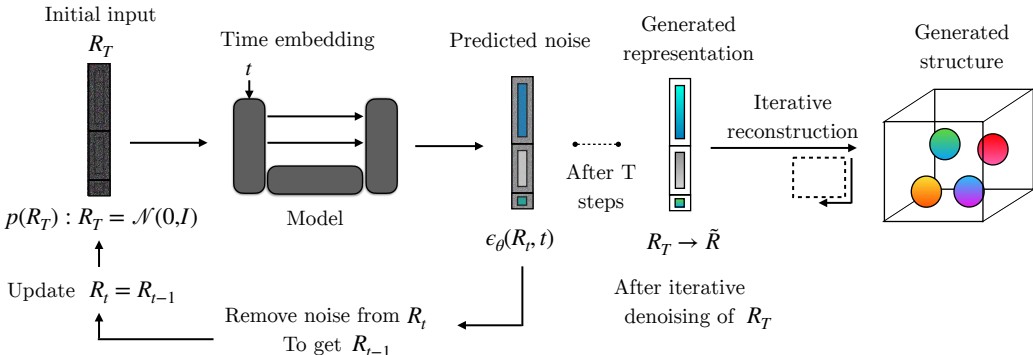

Figure 3: Schematic of the overall generation process of StrucRepDiff. Starting from noisy representation input $\boldsymbol{R}_T$ sampled from $\mathcal{N}(0, I)$, to generating representation $\tilde{\boldsymbol{R}}$ after the denoising process. Finally performing iterative reconstruction of $\tilde{\boldsymbol{R}}$ to get the generated structure.

### 4.3 ITERATIVE RECONSTRUCTION FROM REPRESENTATION SPACE TO CARTESIAN SPACE

Given a generated representation $\tilde{\boldsymbol{R}}$ via the diffusion model, we now reconstruct its corresponding new material structure $\tilde{M}$ in the Cartesian space. Specifically, we can decompose $\tilde{\boldsymbol{R}}$ into $\tilde{\boldsymbol{R}}_{\text{str}}$, $\tilde{\boldsymbol{C}}_{\text{comp}}$ and $\tilde{\boldsymbol{L}}$. We can easily obtain the number of atoms $n$ from $\tilde{\boldsymbol{C}}_{\text{comp}}$ since it is based on normalized frequency encoding. Given $n$ and $\tilde{\boldsymbol{L}}$ and following from the earlier work (Fung et al., 2022), we first randomly initialize a set of Cartesian positions $\boldsymbol{X}$, before calculating the expanded EAD representation $\boldsymbol{R}_{\text{str}}$ based on $\boldsymbol{X}$. Next, we calculate the loss between $\tilde{\boldsymbol{R}}_{\text{str}}$ and $\boldsymbol{R}_{\text{str}}$ that quantifies the difference between the two. Using automatic differentiation to obtain gradients with respect to the atomic positions $\boldsymbol{X}$. These gradients can then be used to update the atomic positions $\boldsymbol{X}$ to minimize the loss between $\tilde{\boldsymbol{R}}_{\text{str}}$ and $\boldsymbol{R}_{\text{str}}$ in an iterative way. The reconstruction process is summarized in Algo. 2. The experimental details and hyper-parameters of the reconstruction algorithm are discussed in appendix A.3.5.

---

**Algorithm 2** Gradient-based structure reconstruction from representation

---

1: **Input:** Generated representation $\tilde{\boldsymbol{R}}_{\text{str}}$; number of atoms $n$ and (optionally) lattice parameters $\tilde{\boldsymbol{L}}$
2: **while** initializations < max_initializations **do**
3:     Randomly initialize a set of Cartesian positions $\boldsymbol{X}$ within the unit cell defined $\tilde{\boldsymbol{L}}$
4:     **while** hops < max_hops **do**
5:         **while** iterations < max_iterations **do**
6:             Calculate $\boldsymbol{R}_{\text{str}}$ based on $\boldsymbol{X}$
7:             Calculate loss: $\mathcal{L}\left(\tilde{\boldsymbol{R}}_{\text{str}}, \boldsymbol{R}_{\text{str}}\right)$
8:             Obtain gradients $\nabla$ from loss using automatic differentiation
9:             Update $\boldsymbol{X}$ to $\boldsymbol{X}'$ with $\nabla$
10:         **end while**
11:         Add Gaussian noise to $\boldsymbol{X}'$ for perturbation.
12:     **end while**
13: **end while**
14: Select $\boldsymbol{X}'$ with the lowest $\mathcal{L}\left(\tilde{\boldsymbol{R}}_{\text{str}}, \boldsymbol{R}_{\text{str}}\right)$
15: Construct crystal structure with $\boldsymbol{X}'$ and $\tilde{\boldsymbol{L}}$

---

## 5 RESULTS

We present the results of this study on a set of well-established benchmarks first proposed by Xie et al. (2021) and later adopted by subsequent materials generation studies. Here we evaluate our ap-

proach on its reconstruction performance and generation performance. Reconstruction performance measures the ability of the method to recover the correct Cartesian positions of a material from its latent space or its representation. The generation performance measures the ability of the method to generate high quality periodic materials, based on the metrics of structural and compositional validity, density and element number statistics, and measures of the coverage. The validity, property, and coverage metrics are further described in appendix A.3.6 In both cases, the performance is evaluated for three distinct, curated datasets of periodic materials. The datasets include (a) the Perov-5 dataset (Castelli et al. (2012)) with perovskite materials. (b) the Carbon-24 dataset (Pickard (2020)) with structures composed entirely of carbon atoms, and (c) the MP-20 dataset (Jain et al. (2013)) with materials displaying diversity in both structure and composition. The purpose of this set of benchmarks is to provide a comprehensive evaluation of the model performance for general materials generation problems.

**Baselines.** We evaluate StructRepDiff by comparing it with several existing material generation methods: FTCP (Ren et al., 2020), Cond-DFC-VAE (Court et al., 2020), G-SchNet (Gebauer et al., 2019), P-G-SchNet(Gebauer et al., 2019) CDVAE (Xie et al., 2021), Symat(Luo et al., 2023) and LM-AC (Flam-Shepherd & Aspuru-Guzik, 2023). FTCP and Cond-DFC-VAE generate materials using Fourier-transformed crystal property matrices and VAE models, respectively. However, Cond-DFC-VAE is limited to generating cubic systems and is only applied to the Perov-5 dataset with cubic structures. G-SchNet and its variant P-G-SchNet are autoregressive 3D molecule generation methods, CDVAE and Symat are periodic material generation method, While LM-AC is the latest language model based material generator. All the methods are compared on the three datasets (except LM-AC which was only performed for the MP and Perovskite dataset). In our experiments, we used a 60-20-20 random split of three datasets: MP-20, Perov-5, and Carbon-24, consistent with previous works.

Table 1: Reconstruction performance of our model based on the reconstruction metric. P, C and MP refers to Perov-5, Carbon-24 and MP-20 datasets respectively. In this context, an upward arrow ↑ indicates that higher metric values correspond to improved performance, while a downward arrow ↓ signifies the opposite. Additional details about the metric and the table are discussed in appendix A.5

| Method | Match rate (%) ↑ | | | RMSE ↓ | | |
|---|---|---|---|---|---|---|
| | **P** | **C** | **MP** | **P** | **C** | **MP** |
| FTCP | 99.34 | 62.28 | 69.89 | 0.0259.71 | 0.2563 | 0.1593 |
| Cond-DFC-VAE | 51.65 | - | - | 0.0217 | - | - |
| CDVAE | 97.51 | 55.22 | 45.43 | **0.0156** | **0.1251** | 0.0356 |
| StructRepDiff | **100.0** | **96.93** | **83.00** | 0.024 | 0.18 | **0.0217** |

First, we observe a good performance for the reconstruction of a material from its representation for our approach in table 1. This is measured using the StructureMatcher method from pymatgen to compare two structures in an invariant manner, and originally used by Xie et al. (2021) and evaluated for FTCP, Cond-DFC-VAE, and CDVAE. Namely, we are able to obtain a near 100% match rate between ground truth and reconstructed structures for the Carbon-24 and Perov-5 dataset, while we get a significant increase in performance for the MP-20 dataset based on the tolerance of stol = 0.5, angle tol = 10, ltol = 0.3, and the lowest RMSE for the MP-20 dataset, while being close in the RMSE value for the other datasets. The metrics for table 1 are discussed in further details in appendix A.5. The significantly better performance compared to FTCP, Cond-DFC-VAE, and CDVAE is likely due to fact that the mapping from structure to representation in our approach is not a trainable function but has an exact analytical form, and that our reconstruction algorithm is effective one at finding the optimum solution. The comparison on methods of reconstruction is discussed in appendix A.4. Having demonstrated that materials generated in our proposed representation space can be successfully decoded back to the correct structure, we move on to its end-to-end generation performance.

Based on the results in table 2, it can also be seen that our model performs competitively with state-of-the-art models like SyMat and the other models in the list, and obtains the best in 6 out of the 16 metrics and comes 2nd best in 4 metrics, supporting the validity of our approach. We also visualize randomly sampled structures for each dataset, and find they closely reflect the characteristics of the

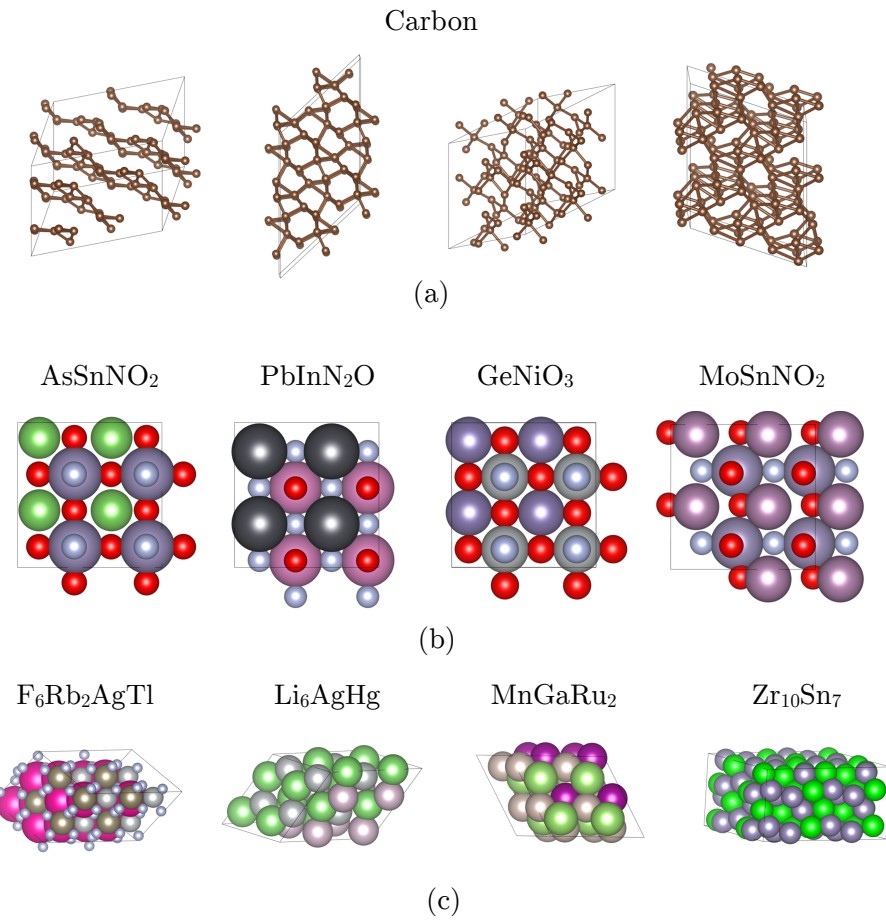

Figure 4: Generated structures from the model. (a) Examples of generated structure from the Carbon-24 dataset.(b) Examples of generated structure from the Perov-5 dataset.(c) Examples of generated structure from the MP-20 dataset.

original training dataset fig. 4. For instance, the generated carbon structures correctly exhibit highly coordinated networks of atoms, and the generated perovskites follow the cubic or distorted cubic of the classic perovskite structures, as well as all having ABX3 compositions where A and B are metal cations, and X is an electronegative anion. For the more diverse MP-20 dataset, we see a similar diversity in the generated structures, while maintaining reasonable bond distances as evidenced by the non-overlapping spheres (based on atomic radii) and plausible compositions containing primarily of two or three unique elements. We also visualize the coverage of the generated samples (in red) compared to the training data (in black) by the plotting first two principal components of the representations of the materials in the dataset. We find the generated samples fully match the distribution of the testing data as shown in fig. 5.

We opted to not perform property optimization or condition generation in this work, owing to the lack of a consistent evaluation method in current studies. In CDVAE and SyMat, separate surrogate models were trained to predict the property being optimized and used the evaluate the generation performance. However, as differently constructed and differently trained surrogate models were used in each study, they do not provide a reliable comparison across the studies, and would consequently be misleading. While a fully first principles method such as density functional theory may be used, they are prohibitively expensive to run.

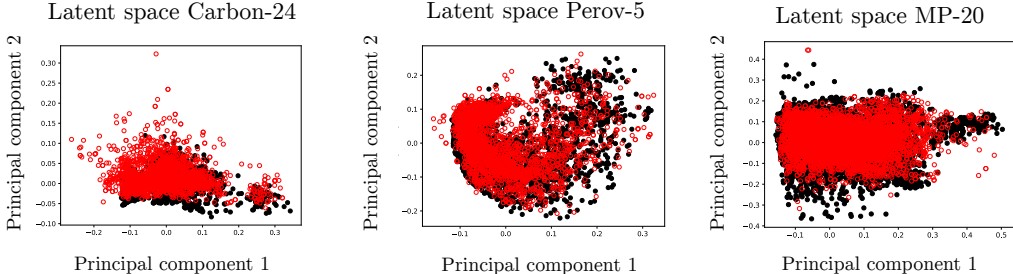

Figure 5: PCA of $R_{EAD}$ of the generated samples (red) overlaid over the training data (black)

Table 2: Generation performance across the Perov-5, Carbon-24, and MP-20 datasets. Here, an upward arrow ↑ indicates that higher metric values correspond to improved performance, while a downward arrow ↓ signifies the opposite. In the results the numbers in bold represent the best performances and (†) represent the second-best performances, respectively.

| Dataset | Method | Validity (%) ↑ | | EMD ↓ | | COV ↑ | |
|---|---|---|---|---|---|---|---|
| | | Comp | Str | # | $\rho$ | R | P |
| **MP-20** | FTCP | 48.37 | 1.55 | 0.7363 | 23.71 | 5.26 | 0.23 |
| | G-SchNet | 75.96 | 99.65† | 0.6411 | 3.034 | 41.68 | 99.65 |
| | P-G-SchNet | 76.40 | 77.51 | 0.6234 | 4.04 | 44.89 | 99.76 |
| | CDVAE | 86.70 | **100.00** | 1.432 | 0.6875 | 99.17 | 99.64 |
| | SyMat | 88.26† | **100.00** | 0.5067 | **0.3805** | 98.97 | **99.97** |
| | LM-AC | **88.87** | 95.81 | 0.092 | 0.696 | 99.60†* | 98.55* |
| | StructRepDiff | 80.52 | 94.16 | **0.081** | 0.673† | **99.67** | 99.79† |
| **Perov-5** | FTCP | 54.24 | 0.24 | 0.629 | 10.27 | 0.00 | 0.00 |
| | G-SchNet | 98.79 | 99.92 | 0.0368 | 1.625 | 0.25 | 0.37 |
| | P-G-SchNet | **99.13** | 79.63 | 0.452 | 0.2755 | 0.56 | 0.41 |
| | CDVAE | 98.59 | **100.00** | 0.0628 | 0.1258† | 99.50 | 98.93 |
| | SyMat | 97.40 | **100.00** | **0.0177** | 0.1893 | 99.68† | 98.64 |
| | LM-AC | 98.79† | **100.00** | 0.028 | **0.089** | 98.78* | 99.36†* |
| | StructRepDiff | 98.17 | 99.99† | 0.059 | 0.289 | **99.84** | **99.79** |
| **Carbon-24** | FTCP | - | 0.08 | - | 5.206 | 0.00 | 0.00 |
| | G-SchNet | - | 99.94† | - | 0.9327 | 0.00 | 0.00 |
| | P-G-SchNet | - | 48.39 | - | 1.533 | 0.00 | 0.00 |
| | CDVAE | - | **100.00** | - | 0.1407 | **100.00** | **99.98** |
| | SyMat | - | **100.00** | - | 0.1195† | **100.00** | 97.59 |
| | StructRepDiff | - | 99.76 | - | **0.1187** | **100.00** | 99.56† |

## 6 CONCLUSION

We introduce StructRepDiff as a new paradigm for the efficient generation of periodic materials which is performed in the representation space. The proposed method is able to perform competitively among all the previous and current methods. This work opens up new possibilities for future exploration for materials generation in the representation rather than Cartesian space, which is currently unexplored. We emphasize here that this approach is theoretically model and does not require any custom architectures or training procedures. Furthermore, it can also be applied to any other suitable, differentiable, representation which exhibits the necessary invariances. This work highlights the benefits of incorporating prior developments from the atomistic modelling domain, namely the use of structural descriptors, to greatly simplify the training and improve the performance of generative models for materials.

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

# A APPENDIX

## A.1 PROOF OF PROPOSITIONS

### A.1.1 PROPOSITION 1 : THE REPRESENTATION VECTOR $\boldsymbol{R}$ OF A MATERIAL $\mathcal{M}$, FOLLOWS ALL THE PROPERTIES OF SYMMETRY OF A PERIODIC CRYSTAL

For any material $\mathcal{M}$ we showed the representation $\boldsymbol{R} = [\boldsymbol{R}_{\text{str}} \oplus \boldsymbol{C}_{\text{comp}} \oplus \boldsymbol{L}]$. We prove the representation vector is invariant to rotation and permutation, translation and periodic transformation.

1.  Invariance to translation and periodic transformation: Given the representaion $\boldsymbol{R} = [\boldsymbol{R}_{\text{str}} \oplus \boldsymbol{C}_{\text{comp}} \oplus \boldsymbol{L} \oplus]$ ; we can represent a structure with a translation as $\boldsymbol{R'} = [\boldsymbol{R'}_{\text{str}}, \boldsymbol{C}_{\text{comp}}, \boldsymbol{L}, \boldsymbol{D}]$, as only $\boldsymbol{R}_{\text{str}}$ is a function of position. From eq. (2) the atomic position dependent term $\phi_i$ remains unchanged as $R_{ij} = |\mathbf{r}_i - \mathbf{r}_j| = |(\mathbf{r}_i + \mathbf{d}) - (\mathbf{r}_i - \mathbf{d})|$ for a translation of the crystal along $\mathbf{d}$. The same argument holds true for periodic transformation, where translation distance $\mathbf{d} = \mathbf{d}_{\text{crystal}}$.

2.  Invariance to rotation: Given the representaion $\boldsymbol{R} = [\boldsymbol{R}_{\text{str}} \oplus \boldsymbol{C}_{\text{comp}} \oplus \boldsymbol{L}]$; We can represent the rotated structure representation as $\boldsymbol{R'} = [\boldsymbol{R'}_{\text{str}} \oplus \boldsymbol{C}_{\text{comp}} \oplus \boldsymbol{L}]$, the rotation of the structure affects the positions $x_i \; \forall i \in [1, N]$ of the $N$ atoms within the crystal. However the relative distances $|\mathbf{r}_i - \mathbf{r}_j|$ between the atoms remains unchanged during any such rotation. Following eq. (2) we observe $\boldsymbol{R'}_{\text{str}} = \boldsymbol{R}_{\text{str}}$.

3.  Invariance to permutation: Given the representaion $\boldsymbol{R} = [\boldsymbol{R}_{\text{str}} \oplus \boldsymbol{C}_{\text{comp}} \oplus \boldsymbol{L}]$, we can permute the atoms within the crystal to obtain an updated representation Given the representation $\boldsymbol{R'} = [\boldsymbol{R'}_{\text{str}} \oplus \boldsymbol{C'}_{\text{comp}} \oplus \boldsymbol{L'}]$. But permuting the atoms we get the same one-hot representation $\boldsymbol{C}_{\text{comp}}$ as the atoms itself don't change. Simillarly, $\boldsymbol{L}$ remains the same as the lengths and angles of the unit cells don't change in magnitude. Finally, for $\boldsymbol{R'}_{\text{str}}$, from eq. (2) the permutation of atoms doesn't change the result $\phi_i$ as the order of summation over all the atoms ($\sum_{i \neq j}^{N} Z_j \Phi(R_{ij})$) is invariant to permutation in the atoms.

## A.2 EMBEDDED ATOM DENSITY

The Embedded Atom Density (EAD) descriptor, inspired by the Embedded Atom Method (EAM) described by Zhang et al. (2019). The EAM theory incorporates a mathematical formulation that accommodates the consistent electron density enveloping each 'embedded' atom, coupled with a short-range nuclear repulsion potential. Furthermore, this representation involves the square of the linear combination of atomic orbital components to adapt to various crystal systems, expressed as the density invariant ($\phi_i$) at the position of atom $i$. eq. (2).

$$\phi_i = \sum_{l_x, l_y, l_z}^{l_x + l_y + l_z = L} \frac{L!}{l_x! l_y! l_z!} \left( \sum_{i \neq j}^{N} Z_j \Phi(R_{ij}) \right)^2 \tag{2}$$

In the above formulation, $Z_{ij}$ denotes the atomic number of a neighboring atom labeled as $j$. $|\mathbf{r}_i - \mathbf{r}_j|$; where $\mathbf{r}_i = \{x_i, y_i, z_i\}$ and $\mathbf{r}_j = \{x_j, y_j, z_j\}$ being the Cartesian coordinate vectors of the central atom $i$ and a neighbor atom $j$ The variable $L$ stands for quantized angular momentum, while $l_{x,y,z}$ signify quantized directional-dependent angular momentum components. Finally $\Phi(R_{ij})$ is a Gaussian-type orbital centered at atom $j$ parameterized by its center $\mu$, width $\eta$ and angular momenta $L = l_x + l_y + l_z$. $f_c$ is a cutoff function continuously damping the invariant to zero at the cutoff radius ($r_c$), and $N_c$ is the number of atoms within $r_c$. The specific mathematical expression for $\Phi$ is as show in eq. (3):

$$\Phi(R_{ij}) = x_{ij}^{l_x} y_{ij}^{l_y} z_{ij}^{l_z} \cdot \exp{-\eta(R_{ij} - \mu)^2} \cdot f_c(R_{ij}) \tag{3}$$

Additionally, the Electron Affinity Difference (EAD) can be viewed as an enhancement of Gaussian symmetry functions. EAD does not differentiate between radial and angular terms. In application the EAD descriptor is a local descriptor for each atom in the material. Thus, to make the overall descriptor for the material permutationally invariant, we pool over all atoms. This approach, which

we propose in this work, has proven to be effective. It allows for efficient information retention without significant loss during reconstruction.

### A.3 EXPERIMENTAL DETAIL

#### A.3.1 DETAILS ABOUT EAD PARAMETERS

In this work, we use $L \leq 1$, $\mu = \{0, 0.1, 0.2, ..., 20\}$, $\eta = \{1, 20, 90\}$, and $r_c = 20$.

#### A.3.2 DETAILS ABOUT ARCHITECTURE AND MODEL

The model for the reverse diffusion process is based on a U-net architecture (Ronneberger et al., 2015), with 1 dimensional convolution blocks (LeCun et al., 1995). The fundamental blocks of our U-net architecture is composed of 2 Res-Net blocks. We implement the Res-Net blocks with weight standardize convolutions followed by Group–Norms as shown by Kolesnikov et al. (2020). We implement 2 fundamental blocks in each downsampling, middle and upsampling stages with a linear attention block between each stage to enhance model's expressivity. The linear attention block is chosen to be a multi-head attention block with 4 attention heads and a 256 dim final embedding size. In the upsampling stage each block is also attached with a skip connection coming from the downsampling stages. Finally a 1D convolution block to the produce the resultant representation vector $\boldsymbol{R}$. The model takes in a batch of noised inputs from the forward diffusion model and predicts noise at various time-steps of diffusion.

#### A.3.3 EXTENDED BACKGROUND ON GENERATIVE DIFFUSION MODEL

In our approach, we leverage the existing denoising diffusion probabilistic models (Ho et al., 2020) with a goal to denoise the representation of the input materials. Diffusion models have been effective in tasks such as image synthesis, denoising, and data imputation; As they evolve data through incremental transformations of random samples (Cao et al., 2022; Croitoru et al., 2023). Starting with the input vector $r_0$, we progressively add Gaussian noise over $T$ steps.

**Forward process.** Given data-point $\boldsymbol{r}_0$ drawn from the actual data distribution $q(\boldsymbol{r})$. In this scenario, it is possible to establish a progressive diffusion process by introducing noise. To be precise, at every Markov chain iteration, we incorporate Gaussian noise with a variance of $\beta$ to $\boldsymbol{r}_{t-1}$, resulting in the creation of a fresh latent variable $\boldsymbol{r}_t$. This new variable $\boldsymbol{x}_t$ follows a distribution $q(\boldsymbol{x}_t|\boldsymbol{x}_{t-1})$.

$$q(\boldsymbol{x}_t|\boldsymbol{x}_{t-1}) = \mathcal{N}(\boldsymbol{x}_t; \mu_t = \sqrt{1-\beta_t}\boldsymbol{x}_{t-1}, \Sigma_t = \beta_t \mathbf{I}) \tag{4}$$

$$q(\boldsymbol{x}_{1:T}|\boldsymbol{x}_0) = \prod_{t=1}^{T} q(\boldsymbol{x}_t|\boldsymbol{x}_{t-1}) \tag{5}$$

If we define $\alpha_t = 1 - \beta_t$, $\bar{\alpha}_t = \prod_{s=0}^{t} \alpha_s$ where $\boldsymbol{\epsilon}_0, ...., \epsilon_{t-2}, \epsilon_{t-1} \sim \mathcal{N}(0, I)$, one can use the reparameterization trick in a recursive manner to obtain the representation vector $\boldsymbol{x}$ at any refinement step $t$ directly from the initial representation vector $\boldsymbol{x}_0$. To produce a sample $\boldsymbol{x}_t$ we can use equation 6.

$$\boldsymbol{x}_t \sim q(\boldsymbol{x}_t|x_0) = \mathcal{N}(\boldsymbol{x}_t; \sqrt{\bar{\alpha}_t}\boldsymbol{x}_0, (1-\bar{\alpha}_t)\mathbf{I}) \tag{6}$$

**Reverse Diffusion.** Generating new structures involves creating them from noisy forward diffusion structures. Learning occurs during the reverse process, where a neural network architecture is used to learn the noise removal process. From the forward process, as $t \to \text{T}$ approaches $\infty$, the latent $\boldsymbol{r}_T$ is nearly an isotropic Gaussian distribution. Therefore if we manage to learn the reverse distribution $q(\boldsymbol{x}_{t1}|\boldsymbol{r}_t)$, we can sample $\boldsymbol{r}_T$ from $N(0, I)$, run the reverse process and get a final sample from $q(\boldsymbol{x}_0)$, generating a novel data point from the original data distribution. The task can be formulated as to approximate $q(\boldsymbol{x}_{t-1}|\boldsymbol{x}_t)$ with a neural network $p_\theta$ as shown in eq. (7), where we take the same functional form of the reverse process as the forward process as shown by Feller (2015). The mean and variance of at each time $t$ is shown as $\mu_\theta(\boldsymbol{x}_t, t)$ and $\Sigma_\theta(\boldsymbol{x}_t, t)$ which are associate different noise levels at different time-step $t$.

$$q(\boldsymbol{x}_{t-1}|\boldsymbol{x}_t) \approx p_\theta(\boldsymbol{x}_{t-1}|\boldsymbol{x}_t) = \mathcal{N}(\boldsymbol{x}_{t-1}; \mu_\theta(\boldsymbol{x}_t, t), \Sigma_\theta(\boldsymbol{x}_t, t)) \tag{7}$$

If we apply the reverse formula for all timesteps $p_\theta(x_{0:T})$, also called trajectory), we can go from $x_T$ to the data distribution. As the reverse process is also sets up as a Markov process from $x_t \rightarrow x_{t-1} \rightarrow ... \rightarrow x_0$. With $p_\theta(\boldsymbol{x}_T) = \mathcal{N}(\boldsymbol{x}_t, 0, I)$ being the pure noise distribution. The aim of the model being, learning $p(\boldsymbol{x}_0)$.

$$p_\theta(\boldsymbol{x}_0) = \int p_\theta(\boldsymbol{x}_{0:T}) d\boldsymbol{x}_{0:T} = \int p_\theta(\boldsymbol{x}_T) \prod_{t=1}^{T} p_\theta(\boldsymbol{x}_{t-1}|\boldsymbol{x}_t) d\boldsymbol{x}_{0:T} \tag{8}$$

**Training.** The objective of the model is to maximise the likelihood of the generated sample $x_0$ to be from the initial data distribution $q(x_0)$. Which in our reverse-process framework would mean maximise the probability $p_\theta(x_0)$, which is the marginalised probability $\int p_\theta(x_{0:T}) dx_{0:T}$ as shown in eq. (8); While learning the parameters $\theta$ of the neural network model. The loss term in eq. (9) takes the simplified form as mentioned by Ho et al. (2020), where we focus on predicting the noise at each step of the diffusion process.

$$\text{Loss} = \mathbb{E}_{(x_0, t, \epsilon)} \left[ ||\epsilon_t - \epsilon_\theta(x, t)||^2 \right] \tag{9}$$

### A.3.4 HYPER-PARAMETERS AND TRAINING DETAILS

---

**Algorithm 3** Diffusion training of the representations

---

1: **repeat**
2:     $R_0 \sim q(R_0) \rightarrow R_0 \in \mathbb{R}^{704}$
3:     $t \sim \text{Uniform}(\{1, 2, ...T\})$
4:     $\epsilon \sim \mathcal{N}(0, I)$
5:     Apply gradient descent on $\nabla_\theta ||\epsilon - \epsilon_\theta(\sqrt{\bar{\alpha}}R_0 + \sqrt{1 - \bar{\alpha}}\epsilon, t)||^2$
6: **until** converged

---

The training of the model was performed on an NVIDIA A100 for a total of 2000 epochs with a learning rate $= 10^{-4}$ batch size of 64. The overall training takes around 12 hrs,6 hrs and 3hrs for MP, Perovskite and Carbon datasets respectively. The number of diffusion steps was set to 2000 following the cosine schedule for addition of the noise. The forward diffusion model destroys the input representaiton using a cosine scheduler from eq. (4) $\beta_t = 1 - \frac{\alpha_t}{\alpha}, 0.990; \frac{\alpha_t}{\alpha_{t-1} = f(t)}$ and $f(t) = \cos(\frac{t/T + s}{1 + s} \cdot \frac{\pi}{2})^2$ as proposed by Dhariwal & Nichol (2021). The hyper-parameters of the model included which are selected based on performance metric explained in appendix A.3.6

1. Architecture based hyper-parameters: Number of Res-net blocks = 2, Attention embedding dimension = 256, Number of attention heads = 4.

2. Diffusion based hyper-parameters: Number of diffusion steps = 2000, Noise scheduler = 'Cosine', Final noise level = 0.999.

3. Training based hyper-parameters: Number of epochs = 2000, Learning rate = $10^{-4}$, Optimizer = Adam, Batch size = 64.

### A.3.5 HYPER-PARAMETERS FOR RECONSTRUCTION ALGORITHM

We implement an L1 Loss for reconstruction, and this loss is used throughout the work. We are using a basin hopping algorithm with a local gradient descent optimization with Adam optimiser.

- The number of basin hopping trials is 6
- The number of basin hops is 7
- The basin hopping step size is a random number in the range [-1, 1]

- The max number of iterations in gradient descent is 300

The loss function values for the iterative reconstruction of our generated structures is usually around [0.005, 0.05] and the reconstruction for errors for valid structures is $\leq 0.01$.

### A.3.6 DETAILS ON COMPARISON METRIC:

In the context of the random generation task, when evaluating COV-R and COV-P metrics, we determine that one material encompasses another material if the disparities in their chemical element composition fingerprints and 3D structure fingerprints fall below certain predefined thresholds, denoted as $\delta_c$ and $\delta_s$, respectively. The structural distance between generated materials and their ground truth counterparts is determined using the Euclidean distance of the CrystalNN fingerprint from Zimmermann & Jain (2020), while the composition distance is assessed through the Euclidean distance of the normalized Magpie fingerprint Ward et al. (2016) .Specifically, for the Perov-5 dataset, we set $\delta_c$ to 6 and $\delta_s$ to 0.8. For the Carbon-24 dataset, we establish $\delta_c$ as 4 and $\delta_s$ as 1.0. Finally, for the MP-24 dataset, we define $\delta_c$ as 12 and $\delta_s$ as 0.6. These values are chosen according to previous publications (Xie et al., 2021; Luo et al., 2023), and we acknowledge that more future work is required to come up with physical explanation towards selection of these thresholds.

1) Validity: According to the criterion proposed by Court et al. (2020) to determine the validity of the generated structures. A structure is considered valid if the shortest distance between any pair of atoms is greater than 0.5 Ångstroms. Additionally, we assess the validity of composition by ensuring that the overall charge is neutral, as calculated by SMACT by Davies et al. (2019).

2) Coverage Metrics (COV): Two coverage metrics, COV-R (Recall) and COV-P (Precision). COV-R measures the percentage of correctly predicted ground truth materials, while COV-P quantifies the percentage of predicted materials with high quality.

3) Property Statistics: To evaluate the property statistics of the generated materials, the Earth Mover's Distance (EMD) between the property distributions of the generated materials and the test materials. The properties considered include density ($\rho$, unit: $g/cm^3$) and the number of unique elements (# elem.).

The mathematical formulation and details of implementation is provided by Xie et al. (2021).

### A.4 MATERIAL RECONSTRUCTION:

The comparison of different methods of reconstruction are summarised for better understanding of the results in section 5. Table 1 measures the ability to reconstruct a material from its representation to the Cartesian positions. In the compared examples, these are from the latent representations which are obtained from a trainable model. FTCP and Cond-DFC-VAE implements some form of variational auto-encoder (VAE) and thus reconstructs the latent vector with the VAE's decoder. While CDVAE reconstructs the material using Annealed Langevin dynamics steps on an initial predicted structure, which is predicted from their VAE's latent space. In our case, our representation can be obtained directly from an analytical function. Nonetheless, the means of evaluation are identical, i.e. we are looking at the Cartesian positions of the reconstructed structure.

### A.5 DETAILS ON RECONSTRUCTION METRICS:

To assess the reconstruction ability of the model, we have employ the StructureMatcher method from Pymatgen Ong et al. (2013) as implemented by Xie et al. (2021). This tool helps identify the closest match between the generated structure and the input structure for all materials in the test dataset, taking into account various material properties. The matching rate is determined by the percentage of materials that meet the specified criteria, which include stol = 0.5, angle tol = 10, and ltol = 0.3. To ensure fairness in our evaluation, the Root Mean Square Error (RMSE) is calculated averaged across all the successfully matched materials. The RMSE used for comparing the original and reconstructed structre is based on Pymatgen StructureMatcher method by Ong et al. (2013). This method calculate RMS displacement between two structures, which is rms displacement normalized by $\left(\frac{\text{Vol}}{\text{nsites}}\right)^{\frac{1}{3}}$ and maximum distance between paired sites only for the structures which pass through the specified match criteria. If no matching lattice is found None is returned.

