# OpenReview forum: "Representation-space diffusion models for generating periodic materials"
_ICLR.cc/2024/Conference — Submitted to ICLR 2024_

### Official Review · Reviewer_FkRz · 2023-10-30

**Soundness:** 2 fair
**Presentation:** 2 fair
**Contribution:** 1 poor
**Rating:** 3
**Confidence:** 5

**Summary:**

The paper described an experimental approach to represent crystal structures with a further aim to guarantee their reconstruction from a given representation.

**Strengths:**

The authors should be highly praised for studying important real objects such as solid crystalline materials.

The paper is generally well-written and contains enough details that helped understand the difficulties.

Figure 1 is especially nice and clearly implies that the required representation (obtained by the first arrow) should be proved to be invertible and realizable (so that any new representation can be realized by a crystal structure) before further stages make sense.

**Weaknesses:**

The word "problem" appears only once (in section 5 on results), though a rigorous and explicit problem statement might have helped the authors to understand the obstacles.

Quote: "The last piece of information needed to fully describe a material structure
M is the lattice parameters L in R^6. Here, we use L directly as the lattice representation"

Comment: This conventional lattice representation (three cell edges and three angles between them) is not invariant because any lattice can be generated by infinitely many different cell bases, for example, the basis (1,0),(n,1) for any integer non-zero n generates the same square lattice.

If we need a complete and invariant description of a periodic crystal, crystallographers solved this problem nearly 100 years ago by using Niggli's reduced cell of a lattice and then recording all atoms in so-called standard settings, see the book "TYPIX standardized data and crystal chemical characterization of inorganic structure types" by Parthé et al, which actually applies to all periodic crystals, not only inorganic.

However, all these standardizations have become obsolete in the new world of big and noisy data because the underlying lattice (not even a unit cell) of any periodic crystal is discontinuous under almost any perturbation, which is obvious already in dimension 1.

For example, the set Z of all integers is nearly identical to a periodic sequence with points 0, 1+ep_1, ..., m+ep_m in the unit cell [0,m+1] for any small ep_1,...,ep_m close to 0, though their minimal periods (or unit cells) 1 and m+1 are arbitrarily different.

This discontinuity was reported for experimental crystals already in 1965, see Lawton SL, Jacobson RA. The reduced cell and its crystallographic applications. Ames Lab., Iowa State Univ. of Science and Tech.

A more recent example from Materials Project shows two nearly identical crystals whose unit cells differ by a factor of (approximately) 2
https://next-gen.materialsproject.org/materials/mp-568619
https://next-gen.materialsproject.org/materials/mp-568656

Theorem 13 in Widdowson et al (MATCH 2022) proved that any cell (not only Niggli's) reduction is discontinuous hence using a cell basis is hopeless.

Moreover, atoms in any material always vibrate above absolute zero temperature, so their positions continuously change. As a result, any crystal structure with fixed atomic coordinates in a database is only a single snapshot of a potentially dynamic object, especially for proteins whose structures are also determined often by crystallization.

Quote: "Invariance to permutation: Given the representation R = [Rstr,Ccomp,L], we can permute the atoms within the crystal to obtain an updated representation".

Comment: since a periodic crystal contains infinitely many points, the updated representation will be infinite. Even if only m atoms in a minimal cell are permuted, the representation blows up by the exponential factor of m!, which is also impractical.

Since the paper used PCA in Figure 5, the authors might be interested in learning that any dimensionality reduction is either discontinuous (makes close point distant) or projects an unbounded domain to a single point (loses an infinite amount of data), see the proof in Landweber et al "On Fiber Diameters of Continuous Maps", the American Mathematical Monthly, v.123 (2016), p.392-397.

Quote: "there is no single existing representation that fulfills all three of the conditions mentioned above".

Comment: there are at least two stronger representations that practically fulfill all three conditions and two more important practical requirements of polynomial-time computability and Lipschitz continuity under small perturbations of atoms, see isosets in Anosova et al (DGMM 2021, extended in arxiv:2205.15298) and Pointwise Distance Distributions in Widdowson et al (NeurIPS 2022, extended in arxiv:2108.04798).

In conclusion, the proposed invariants don't satisfy the practical requirements that were fulfilled by the simpler, stronger, and faster invariants.

**Questions:**

How many terms appear in the external sum of formula (2) in appendix A.2?

Does a manually chosen cutoff radius (in appendix A.2) make any invariant incomplete because perturbing atoms can scale up a unit cell to a size larger than any fixed radius?

What is RMSE used for comparing an original and reconstructed structures, and does this RMSE satisfy the metric axioms (en.wikipedia.org/wiki/Metric_space)? If the first axiom fails, distance 0 doesn't guarantee that structures are equivalent. If the triangle inequality fails, the paper by Rass et al (arxiv:2211.03674 ) proves that the popular clustering algorithms such as k-means and DBSCAN can output any predetermined clusters.

How many hidden parameters and CPU hours were used for producing the experimental results?

Did the authors know about the classical results in crystallography cited above, starting from Niggli (1927), Lawton (1965), and Parthe (1987)?

---

> ### Author Response · Authors · 2023-11-20
>
> We are grateful for the reviewer's efforts in improving the quality of this paper. We are grateful to the reviewer for highlighting the strengths of our paper, and more importantly the weaknesses of our work. The following are the detailed response for each of the weaknesses and the question and the changes which are made in the updated submission.
>
> 1. We sincerely acknowledge the knowledge shared by the reviewer which can definitely benefit the development of applied material science. In our work we used the EAD descriptors for our work as it has already been proven to be invariant by Zhang et al. in their work "Embedded atom neural network potentials: Efficient and accurate machine learning with a physically inspired representation." The main objective of our work as correctly mentioned by the reviewer is focused on generating novel materials in the representation space, which we achieve with the competitive performance according to the defined metrics.
>
> 2. The number of terms depends on the value of the angular quantum lumber $L$ chosen, which is a hyper-parameter.
>
> 3. Perturbing an atom beyond the cutoff radius will indeed have no effect on the representation. However, as we pool the local representations for each atom to form a global representation, these changes can still be tracked, which allows us to reconstruct, even if neighbours are locally beyond the cutoff for certain atoms.
>
> 4. The RMSE used for comparing the original and reconstructed structure is based on Pymatgen's `StructureMatcher` method. This method calculates RMS displacement between two structures, If no matching lattice is found, `None` is returned. We have included this in the appendix section A.4.
>
> 5. We have tried to mention all the hyper-parameters in section A.3.1 and A.3.2 for both our architecture and diffusion model. The number of CPU hours required for generating the experimental structures from our generated structure representation was around 12 hours for 10K structures while working with 12 parallel nodes.
>
> 6. We thank the reviewer for mentioning these classic texts in crystallography. We are certain they will prove to be very useful in improving our work.

---

> > ### Comment · Reviewer_FkRz · 2023-11-20
> >
> > >In our work we used the EAD descriptors for our work as it has already been proven to be invariant by Zhang et al. in their work "Embedded atom neural network potentials: Efficient and accurate machine learning with a physically inspired representation."
> >
> > This paper at https://doi.org/10.1021/acs.jpclett.9b02037 studied the EAM descriptors in the finite case, not periodic. Here is the key quote: "in the EAM framework, the total energy of an N atom system is just the sum over all individual impurity energies". The acronym EAD didn't appear in this past work, while the word "lattice" was mentioned only once. Moreover, formula (2) in the recommended paper involved factors of the form x^{l_x} y^{l_y} z^{l_z}. Are they preserved by any SO(3)-rotation on x,y,z?

---

> ### Author Response · Authors · 2023-11-21
>
> The EAD descriptor we describe refers to the implementation from the paper by Zhang et al. [1] We followed the naming convention used by a following work [2], which is why the name does not show up in the original paper. This is as they describe, inspired by the EAM model, but it is not the same as the EAM model.
>
> In the paper by Zhang et al, they describe the descriptor (which we refer to as EAD) as being invariant to translation, rotations and permutations. This descriptor is also successfully applied to periodic systems such as bulk metals, oxides, and alloys.
>
> ## References
>
> [1]: [Zhang, Yaolong, Ce Hu, and Bin Jiang. "Embedded atom neural network potentials: Efficient and accurate machine learning with a physically inspired representation." The journal of physical chemistry letters 10.17 (2019): 4962-4967.](https://pubs.acs.org/doi/abs/10.1021/acs.jpclett.9b02037)
>
> [2]: [Yanxon, Howard, et al. "PyXtal_FF: a python library for automated force field generation." Machine Learning: Science and Technology 2.2 (2020): 027001.](https://iopscience.iop.org/article/10.1088/2632-2153/abc940/meta)

---

### Official Review · Reviewer_gu8x · 2023-10-31

**Soundness:** 2 fair
**Presentation:** 2 fair
**Contribution:** 2 fair
**Rating:** 5
**Confidence:** 3

**Summary:**

This paper aims at building a tool for generating periodic materials (molecules represented by atom types, their coordinates, and the lattice structure). The key idea of the paper is to perform diffusion-based generative modeling on a tailored representation space. The representation space is a result of concatenation of a physically meaningful representation, i.e., Embedded Atom Density (EAD), with explicit information extractable from the coordinate based representation, i.e., composition and lattice parameters. A key step in generating molecules with explicit representations (atom coordinates) is the reconstruction from the proposed representation space. The paper mentions that this is the first use of latents (in contrast to coordinate-based representations) for generating periodic materials. The benefit of the proposed method is that the generated molecules are invariant to different transforms. The results show acceptable performance in terms of reconstruction step as well as generation metrics, such as validity.

**Strengths:**

- The paper seems to be original in the scope of problem definition. Although the question of molecular generation is currently being widely studied, the paper positions itself around periodic materials. From the Related Work section, it seems that this area has not received enough attention. In terms of methodology, the paper relies considerably on prior work for generation, representation, and reconstruction steps.

**Weaknesses:**

- The technical contribution, while considerable, is not deep enough for a venue like ICLR. Although there is nothing wrong with combining existing methods, for this kind of contributions one expects a set of deeper insights or surprisingly better results than state of the art.

- The paper could have been better exposed. For example, it is not clear if the proposed concatenated representation is the idea of this paper or not. I assume it is, but have not seen the paper mention it explicitly. In general, mentioning an explicit list of contributions is recommended. Another example is the reconstruction procedure (section 4.3). While it is a significant part of the proposed method, it is only later in the results section that we understand this step is done through back-propagation using automatic differentiation of an analytical function.

- The paper tries to evaluate its proposed representation in the context of generation. This is Ok. But if the concatenated representation is the key, the focus of evaluation should have been that part. For example, an extensive evaluation of different invariances especially the permutation invariance.

- It is true that running density functional theory might be time consuming, but I think this is necessary for the evaluation. Obviously, the evaluation results based only on surrogate models cannot be judged clearly.

**Questions:**

- What is the motivation behind min-, max-pooling of the EAD representation? Is this known or the paper proposes it?

- Is Table 1 a fair evaluation? That means, do the competing methods in the table have a dedicated component for the reconstruction phase (like the method of Fung et al. which the paper relies on)?

- Table 1: For generating this table, have the valid molecules first been separated for all methods? Or only for a few of them? If only for a few of them, which ones?

---

> ### Author Response · Authors · 2023-11-20
>
> We are grateful for the reviewer's efforts in improving the quality of this paper. We are grateful to the reviewer for highlighting the strengths of our paper, and more importantly the weaknesses of our work. The following are the detailed response for each of the weaknesses and the question and the changes which are made in the updated submission.
>
> 1. The EAD descriptor is a local descriptor for each atom in the material. To make the overall descriptor for the material permutationally invariant, we pool over all atoms. This approach, which we propose in this work, has proven to be effective. It allows for efficient information retention without significant loss during reconstruction
>
> 2. Table 1 measures the ability to reconstruct a material from its representation to the Cartesian positions. In the compared examples, these are latent representations which are obtained from a trainable model. In our case, our representation can be obtained directly from an analytical function. Nonetheless, the means of evaluation are identical, i.e. we are looking at the Cartesian positions of the reconstructed structure.
>
> 3. The reviewer is correct in pointing at the separation of valid materials before the evaluation. As mentioned in our appendix section A.4 The Match rate is computed for materials which lie within the tolerance limits of ltol = fractional length tolerance, stol = site tolerance, atol = angle tolerance as per pymatgen's StructureMacher method. While the RMSE values are calculated over all successful matches only. Which is effectively calculating the fingerprint distance between a original structure and generated structure which have already passed the cut-off of match. This follows the identical procedure in previous works.
>
> 4. We appreciate your recognition of the technical contributions we have made, even as we understand your perspective regarding the depth required for a venue like ICLR. We also agree that venue often demands a substantial improvement over the current state of the art. But In our research, we've demonstrated the feasibility of generating valid periodic structures with a method of diffusion in the representation space, a method which has never been tried before. Moreover, we've successfully managed to reconstruct these structures, transitioning them from the representation space to Cartesian space. Judging by the metrics of reconstruction and generation, we are confident about the competitiveness of our model's performance. We recognise that this is just a step in an ongoing journey of discovery and improvement in this field, but an important step towards opening a new paradigm of representation based generative model in the field of material discovery.
>
> 5. We acknowledge the recommendations given by the reviewer for exposing the novelty of the paper in our updated submission on Page 2 in the introduction section. Although our work does involve the approach to represent a concatenated representation of the material in a novel way, but our main objective of this work lies in presenting a novel approach for materials generation by overcoming the constraints of existing representation-based methods, with the help of generative diffusion models.
>
> 6. The inclusion of density functional theory calculations will certainly provide a better platform for validation. However, as mentioned by the reviewer, this method is computationally very expensive and non-trivial to perform. More importantly, it detracts from our model's objectives, which, as previously mentioned, are geared towards pioneering a new paradigm of representation-based generative models in the field of material discovery.

---

> > ### Comment · Reviewer_gu8x · 2023-11-22
> >
> > Thank you for your answer to my review.
> > 1. The pooling is now more clear. It could be highlighted as a small contribution.
> > 2. I understand. Authors could highlight this difference and briefly mention how the reconstruction is done for competing methods.
> > 3. Now is more clear. I think this could also be explained more in the paper.
> > 5. I still recommend having an explicit list of the novel contributions and putting them in perspective of the current work. I think it is perfectly fine to say the reconstruction step is done by [x] and we leveraged it here.
> > 6. I disagree. In my experience, running some of the available DFT software, such as PSI4 (https://github.com/psi4/psi4)  needs less than a couple of days.
> >
> > I thank again the authors for their effort with the paper and with rebuttal. My evaluation remains the same and I hope the authors find this review cycle helpful.

---

> > > ### Author Response · Authors · 2023-11-22
> > >
> > > We thank the reviewer for further important feedback, the following are the detailed responses for each of the concerns and questions, and the changes which are made in the updated submission.
> > >
> > > 1. We thank the reviewer for this insightful suggestion. We have included this in section A.2
> > > 2. We acknowledge the suggestions by the reviewer by creating a detailed section A.4 highlighting the methods used in all respective works of Table 1.
> > > 3. We acknowledge the suggestion and add the details in section A.5
> > > 4. We thank the author for this very important suggestion and acknowledge his recommendations by including an explicit list of our contributions in the Introduction section, with the heading Major contributions on Page 2.
> > > 5. We acknowledge the response by the reviewer and would like to humbly respond to this concern. The mentioned software PSI4 is not a suitable software for periodic systems, and DFT calculations for periodic crystals generally take more time than similarly sized molecular systems. We do not believe DFT evaluations on a small amount of structures within a limited computation budget would be meaningful or provide any additional insights beyond the existing metrics, especially as the prior work being compared against also do not have DFT. A more complete study of the structures with DFT would be more suitable for a subsequent study.
> > >
> > > We would like to extend our sincere thanks to the reviewer once again for their valuable efforts and the detailed, insightful comments provided on our paper. We humbly request that the reviewer kindly reconsider our revised submission, taking into account the explanations and amendments we have made in response to their feedback.

---

### Official Review · Reviewer_iuNS · 2023-10-31

**Soundness:** 3 good
**Presentation:** 2 fair
**Contribution:** 2 fair
**Rating:** 6
**Confidence:** 2

**Summary:**

In this manuscript, the authors explore a methodology for generating periodic materials by combining representation embedding with a diffusion model. Their approach involves the initial projection of the original material structure into a representation space that integrates the Embedded Atom Density (EAD), composition, and lattice representations. Subsequently, a noise and denoising diffusion process is applied to the embedded representation, and the material structure is reconstructed from this representation.

**Strengths:**

- Generating periodic materials while preserving their structures is essential for accurately capturing and reconstructing the material's structural properties. The authors address this challenge by integrating representation embedding and the diffusion process. The way is straightforward and powerful.

- The manuscript provides a comprehensive description of the model algorithms and experimental methods, ensuring a clear understanding of their model's computations.

**Weaknesses:**

- In the sections of Introduction and Related Works, the generation methods are discussed in terms of representation-based generation and direct material generation. However, when the authors describe the baseline models in the Results section, the two categories are not explicitly mentioned, leading to ambiguity regarding the categorization of these models. It appears that the authors exclusively compared their methods with those of direct material generation. This raises questions about the extent to which the current representation embedding solely contributes to the improvement and to what extent the diffusion process is necessary for their proposed methodology.


- I’m not familiar with the datasets the authors used, but I wonder how the test (or validation) data is divided from the training data. For example, the authors evaluate the PCA distribution analysis using the “training” data in Figure 5 according to the main text (page 7), which should be analyzed using the test data.


- The authors mention “significant increase” and “significantly better” in the results section (page 7). Are these based on some statistical tests? If so, the authors should describe the details.


- Figure 1’s terminologies are not consistent in the 4.1. The descriptions of Cell, Comp, and Structure would be better matched with those in the main text, at least in the caption of Figure 1. Besides, when the authors explain Section 4.1, referring to Figure 1 would be easier to understand.


- At first, I thought that the hyperparameter of training was not described in the manuscript. Later, I learned it’s described in Appendix 3.4.4. Including a reference to Appendix 3.4.4 in the main text would help readers’ understanding.


- The reference of algo. 2 on page 5 should be algo. 1.


- The source code is not available in the supplementary material. Sharing the code will contribute to future development in the field.

**Questions:**

- Let me confirm that the component distribution of generated and original samples in Figure 5 is computed from the same PCA loadings and these axes are directly comparable. I wonder why the PCA distribution of generated samples (Figure 5 left in particular) is more diverse than the original distribution.

- Could the authors explain more why the RMSE results are not improved much compared to those of the Match Rate?

---

> ### Author Response · Authors · 2023-11-20
>
> We are grateful for the reviewer's efforts in improving the quality of this paper. We are grateful to the reviewer for highlighting the strengths of our paper, and more importantly the weaknesses of our work. The following are the detailed response for each of the weaknesses and the question and the changes which are made in the updated submission.
>
> 1. Yes, it is correct that the axes are directly comparable for the generated and the test samples.
>
> 2. The reviewer is correct in noting the slightly higher diversity in the distribution of the generated structures. This is likely due to problems in the reconstruction step from representation to the Cartesian positions. As it is possible for the reconstruction to not fully converge (due to us choosing a fixed number of iterations), the partially reconstructed structures may not well represent the training distribution. We note that the number of these outliers is proportionally small, and can be easily screened out by removing structures with a high reconstruction loss.
>
> 3. The reviewer correctly observes less improved RMSE results as compared to the Match Rate. As mentioned in our apendix section A.4 The Match rate is computed for materials which lie within the tolerance limits of ltol = fractional length tolerance, stol = site tolerance, atol = angle tolerance as per pymatgen's StructureMacher method. Meanwhile the RMSE values are calculated over all successful matches only. Which is effectively calculating the fingerprint distance between a original structure and generated structure which have already passed the cutt-off of match. Hence, even lower match rates can give certain structures which are closer in fingerprint distance, while higher match rate may give a higher RMSE value along. As can be seen in the case of CDVAE's C and MP RMSE columns.
>
> 4. The reviewer correctly mentions the comparison of our model with direct material generation, as the current models have not implemented generative model in the representation space. And which brings the novelty to our approach. Nevertheless, we have included the work done by Uhrin (2021) [1] in section 2.1 of our paper, while the study does highlight about the possible use of generative models in the representation space but did not integrate these representations into a generative model, which somewhat limits their immediate applicability and hence no comparative analysis has been presented with this model.
>
> 5. Yes, the reviewer correctly observes the typing error on page 7 regarding figure 5 of our paper. All the three subfigures of figure 5 have been analysed on the testing dataset. We humbly acknowledge the mistake and have updated this in the submission.
>
> 6. Certainly, the significant increase in performance for the metrics mentioned in Table 1 are related to the Match rate (%) and RMSE (in the case of comparison for the MP dataset). The insightful suggestion on details for these metrics have been humbly acknowledged in the appendix section A.4 of the paper.
>
> 7. The insightful suggestion to improve Figure 1's caption for better understanding of the context has been acknowledged in the updated submission. The updated caption will indeed help the reviewers get better insight of the process.
>
> 8. The suggestion of including the reference of Appendix with hyper-parameters information has been acknowledged. Moreover, the reference to Algo 2 in section 4.3 on Page 5 is indeed correct, as there we are mentioning about our iterative reconstruction process.
>
> 9. Certainly, the source code for the model will benefit future development in the field, and we highly support this suggestion. However, aligning with the instructions of the initial submission, we have not yet included the link to our source code. We plan to make it publicly available as soon as the review process is finished.
>
> ## References
>
> [1]: [Uhrin, Martin. "Through the eyes of a descriptor: Constructing complete, invertible descriptions of atomic environments." Physical Review B 104.14 (2021): 144110.](https://journals.aps.org/prb/abstract/10.1103/PhysRevB.104.144110)

---

> > ### Comment · Reviewer_iuNS · 2023-11-22
> > **Responses to Authors**
> >
> > I appreciate the responses from the authors. My concerns were addressed in the authors' reply.

---

### Official Review · Reviewer_dKZ8 · 2023-11-04

**Soundness:** 3 good
**Presentation:** 3 good
**Contribution:** 1 poor
**Rating:** 3
**Confidence:** 4

**Summary:**

The work proposes a generative modeling of materials structures. It first encodes the structure into a symmetry-preserving representation and then trains a generative model on this representation, thereby bypassing the problem of respecting a set of symmetries (euclidean + permutation), as has to be done in direct generation. It then measures the performance on an existing benchmarks of crystal materials structures.

**Strengths:**

The idea of dealing with symmetries by learning in a controllable deterministic invariant space is nice (albeit not novel, see below). The paper clearly outlines the main ideas. The implementation seems well done.

**Weaknesses:**

It seems pretty clear this paper is far from finished. There's an idea, there's some early results, but this really just needs more work. It's impossible to tell at this stage if this idea work (it may, we just cannot tell).

In particular, there are two key weaknesses

- The evaluations are not very meaningful. The paper sets out to improve materials discovery through generative modeling. The goal of materials discovery is to find materials that are a) novel, b) thermodynamically stable, c) have interesting properties. The paper addresses only point a), and only to some extent. There isn't a single DFT calculation run to see if any of these materials would be stable? The word "stable" isn't even mentioned. That means it may be that all of these predicted structures could never exist in nature. Similarly, there's zero addressing of whether these (potentially unstable) materials have useful properties. Without addressing this, it's just impossible to assess if this is a good idea or not.

- The paper would benefit from discussing the relationship to Uhrin 2021, "Through the eyes of a descriptor: Constructing complete, invertible descriptions of atomic environments"  https://arxiv.org/pdf/2104.09319.pdf -- which describes many of the core ideas discussed here.

**Questions:**

None.

---

> ### Author Response · Authors · 2023-11-20
>
> We are grateful for the reviewer's efforts in improving the quality of this paper. We are grateful to the reviewer for highlighting the strengths of our paper, and more importantly the weaknesses of our work. The following are the detailed response for each of the weaknesses and the changes which are made in the updated submission.
>
> 1. We disagree with this assessment as we have shown a complete end to end example of structure generation following this concept, and have provided the same evaluations and metrics used by all prior studies investigating structure generation. We also ask the reviewer to provide a more meaningful definition of if the idea "works." The problem of materials discovery (with generative modelling) is a challenging one, and it is difficult to expect that all problems associated with it can be resolved in a single study. What we have done here is shown the viability of generating valid periodic structures using a standard generative model, and reconstructing the generated structures from the representation space back to the Cartesian space. Based on the reconstruction and generation metrics, we believe we have been successful in this regard. We admit that there is still significant work which must be done in this area, but this requires a concerted effort in the community to develop better metrics, datasets, and validation with computational simulations.
>
> 2. We note that these evaluations have been used extensively in prior studies and allow us to provide a fair comparison between our method and these prior studies. There are inevitably limitations in these metrics , but it is not reasonable to discount a growing field and community in generative modelling for atomic structure for lack of better metrics.
>
> 3. We acknowledge the suggestion by the reviewer in our updated submission and are thankful for the suggestion. Although, we would like to humbly suggest the following key areas where our work moves forward from Uhrin (2021) [1]
>
>     - The paper does show the use of descriptors in representing structures and iteratively reconstructing back to structures, but has shown no instances of application to a generative model. As such it is impossible to evaluate if it is a successful concept or not in the context of generative modelling beyond simple speculation.
>     - The authors propose a specific descriptor and iterative procedure for reconstruction which works in a similar concept, but our approach is valid for any differentiable descriptor, which goes beyond the specific descriptor proposed in the paper.
>
>
> ## References
> [1]: Uhrin, Martin. "Through the eyes of a descriptor: Constructing complete, invertible descriptions of atomic environments." Physical Review B 104.14 (2021): 144110.

---

> > ### Comment · Reviewer_dKZ8 · 2023-11-21
> >
> > I acknowledge that I have read the response. My scores remain unchanged as the weaknesses remain unaddressed.

---

### Official Review · Reviewer_myBa · 2023-11-05

**Soundness:** 1 poor
**Presentation:** 3 good
**Contribution:** 2 fair
**Rating:** 6
**Confidence:** 4

**Summary:**

This work proposes StructRepDiff, a novel diffusion-based model for 3D periodic material generation. StructRepDiff uses a denoising diffusion model to generate periodic materials in the form of symmetry-invariant representations, including compsition representation, lattice parameters, and embedded atom density representations. Atom coordinates are obtained by searching coordinates that best match the generated embedded atom density representations with gradient descent. Experiments show that StructRepDiff performs well in reconstruction and random generation of periodic materials.

**Strengths:**

- **Originality**: Several novel ideas are proposed by this work, including generating materials in the form of symmetry-invariant representations and searching atom coordinates to match embedded atom density representations.
- **Quality**: Generally, the key points of the proposed method are clearly described, and the proof that the used material representations are invariant to rotation, translation, and periodic transformations is given in the appendix. Experiments on benchmark datasets show the proposed model achieves good performance in periodic material generation.
- **Clarity**: The writing of this paper is clear, well-organized and easy-to-follow.
- **Significance**: The proposed model is useful for novel material design in a broad range of real-world applications.

**Weaknesses:**

- It is not clear how the number of atoms $n$ is obtained from $C_{comp}$ as $C_{comp}$ is normalized by $n$. It is possible that materials with different $n$ may have the same $C_{comp}$. For instance, in $C_{comp}$ of materials with only carbon atoms, only the item for carbon atom is 1 while all other items are zeros, but there may exist different number of atoms $n$ in a unit cell. Authors are encouraged to clarify how $n$ is obtained.
- There lack many significant details in the presentation of obtaining coordinates from embedded atom density representations in Section 4.3. Authors are recommended to clarify the following details: (1) What is the formula of loss function $L$? (2) What are the gradient descent algorithm, step size, max_initializations, max_hops, and max_iterations used in experiments? (3) What are general range of the loss function values when optimization ends? It would be better to report the mean final loss function values averaged over all generated materials.
- Table 1 needs  improvements. The column "Dataset" should be removed. Italic font type should not be used to the "P" under "RMSE" to keep consistency.
- Property optimization is important in real-world problems because real-world applications usually require to generate materials with some target chemical properties. However, it remains unknown how to do property optimization with the proposed model. Authors are encouraged to discuss how to property optimization with StructRepDiff, and explain whether the failure of achieving invertible mapping between material representations and atom coordinates has negative impacts on property optimization.
- In the fourth line of Section 4.3, authors inappropriately cites their own work as "following from our earlier work (Fung et al., 2022)". This leaks the identities of authors and seems to violate the anonymous policies of ICLR.

------------------Post Rebuttal----------------------
Authors have well addressed all my concerns and questions in their rebuttal. I have raised my score to 6.

**Questions:**

In addition to embedded atom density representations, are there any other ways for 3D structure representations? Authors are encouraged to give some alternative examples for embedded atom density representations.

---

> ### Author Response · Authors · 2023-11-20
>
> We are grateful for the reviewer's efforts in improving the quality of this paper. We are grateful to the reviewer for highlighting the strengths of our paper, and more importantly the weaknesses of our work. The following are the detailed response for each of the weaknesses and the question and the changes which are made in the updated submission.
>
> 1. We included a quantity \( n \) which is the total number of atom in the material, concatenated to composition vector. So in generating the materials representation, the number of atoms in the material can be obtained explicitly.
>
>    The reviewer is correct in saying $C_\{comp\}$ for the example both $\( C_6 \)$ and $\( C_8 \)$ will have the same  $C_\{comp\}$, but it will have different values of \( n \) so these materials can still be differentiated in the overall composition vector.
>
> 2. We find a simple $L1$ Loss is sufficient for reconstruction, and this loss is used throughout the work.
>
>    We are using a basin hopping algorithm with a local gradient descent optimization. The optimizer used is Adam.
>
>    - The number of basin hopping trials is 6
>    - The number of basin hops is 7
>    - The basin hopping step size is a random number in the range \[-1, 1\]
>    - The max number of iterations in gradient descent is 300
>
>    The loss function values for the iterative reconstruction of our generated structures is usually around \[0.005, 0.05\] and the reconstruction for errors for valid structures is \( $\leq 0.01$ \). The details have now been added in appendix A.3.5.
>
> 3. The Table 1 has been updated as per the helpful recommendations of the reviewer.
>
> 4. The reviewer is absolutely correct about the importance of property optimisation in this domain of work. However, in the scope of this current work, we aimed to address the implementation and working of our overall method in non-conditionally generating periodic structures which the model is able to do as per our competitive performance from Table 2. We emphasize primarily on the viability of structure generation in our proposed representation space, which is completely novel compared to other current approaches. The inclusion of conditional generation will invariably also require the use of density functional theory calculations for validation, which is a computationally expensive task and non-trivial to do, and detracts from the primary focus of this paper. It is our opinion that evaluating the property prediction with another surrogate model, such as a GNN, is not sufficiently accurate to provide used metrics, nor can they be compared to prior studies in the literature which use different surrogate values. This remains an important problem which needs to be addressed by the community.
>
> 5. Yes, apart from embedded atom density representation, there exists a host of other representations which can be chosen, such as the weighted version of the classic atom centered symmetry function. As we note in this recent review (Chem. Rev. 2021, 121, 16, 9759–9815), there are many other potential options to choose which we do not exhaustively show here. We mainly chose the embedded atom density representation due to having slightly better reconstruction performance over the atom centered symmetry function in our experiments, and a faster evaluation speed. However, this approach is notable for being applicable to any differentiable representation.

---

> > ### Comment · Reviewer_myBa · 2023-11-20
> > **Follow-up Response**
> >
> > I appreciate authors' work in addressing my questions and concerns.
> >
> > - As the total number of atoms is also included in the material representation, please clarify it in Section 4.1.
> > - For the L1 loss used in the reconstruction, are fractional coordinates or Cartesian coordinates used as inputs to the loss function?
> > - In Section 5 and Table 1, the method name "Cond-DFC-VAE" is mistakenly written as "Cond-DFC" for many times. Please correct them.
> > - For property optimization, I understand that reliable experimental evaluation may not be easy, but I am wondering how to property optimization with StructRepDiff and whether the failure of achieving invertible mapping between material representations and atom coordinates has negative impacts on property optimization. I hope authors can give an in-depth discussion about these problems.

---

> ### Author Response · Authors · 2023-11-21
>
> We thank the reviewer for further important feedback, the following are the detailed responses for each of the concerns and questions, and the changes which are made in the updated submission.
>
> 1. We thank the reviewer for the suggestion, following which we have updated our submission's section 4.1
> 2. The L1 loss in our reconstruction algorithm (Algorithm 2) uses the representation of the structures, and not the coordinates directly.
> 3. The reviewer correctly mentions the typing errors regarding the "Cond-DFC-VAE" method. We humbly acknowledge the mistake and have updated this in the submission.
> 4. Property optimization can be performed in our framework by incorporating conditional generation into our diffusion model, without changing any other components of the process. Materials are then generated with the diffusion model conditional on a target property value, and reconstructed to the atomic structure as before. In the future, we would like to evaluate the effectiveness of this approach by performing conditional generation on a target band gap or formation energy, and performing DFT calculations on the structure with the identical settings as the training data from Materials Project. With respect to the impact of the "failure of achieving invertible mapping between material representations and atom coordinates" on property optimization, even though these representations do not have a closed-form solution for inverting to cartesian positions, our iterative reconstruction algorithm is nevertheless able to reconstruct structures very accurately, as measured by the reconstruction metrics in Table 1. Therefore, it is not anticipated to be a critical obstacle in performing property optimization with this approach, especially if the reconstruct is able to recover the positions exactly or near-exactly.

---

> > ### Comment · Reviewer_myBa · 2023-11-21
> > **Follow-up Response**
> >
> > Thank you for your response. I have not seen any updates about including the total number of atoms in material representation in Section 4.1, and the representation dimension is still described as $2N_e+|\mathbb{A}|+7$, which should be $2N_e+|\mathbb{A}|+8$ if including the total number of atoms. Please update Section 4.1 accordingly.

---

> > > ### Author Response · Authors · 2023-11-21
> > >
> > > We thank the reviewer for further important feedback on our response. We have updated Section 4.1 as per the recommended corrections and suggestions.
> > >
> > > 1. The composition vector $C_{\text{comp}}$ has been updated with the concatenated $n$ , i.e the total number of atoms in material $M$. $C_{\text{comp}} = [\hat{C}_{\text{comp}} \oplus n ]$,
> > > - where $\hat{C}_{\text{comp}}$  being the initial normalised composition vector.
> > > 2. The full structure representation ${R} \in \mathbb{R}^N$, where $N = 2N_{{e}}+|\mathbb{A}|+8$
> > > - with $N_{\{e}}$ being the hyper-parameter for the EAD representations and $|\mathbb{A}|=96$, the number of unique elements in our training dataset

---

### Author Response · Authors · 2023-11-22

Dear reviewers,

Thank you for your valuable feedback. We believe it has improved our submission. Given that the author interaction window is coming to an end, we would like to kindly ask you to consider the rebuttal responses and let us know if they address your concerns and, if they do, we would humbly request you to consider raising your scores.

It would be our pleasure to answer any further questions.

---

### Meta-Review · Area_Chair_QsKv · 2023-12-08

**Metareview:**

This paper presents a representation (latent) space diffusion model for generating novel materials. While the reviewers have acknowledged the importance of the problems and merits in this submission, they have not found the main novelties of the submission substantial. There were also concerns about missing DFT analysis. Given these concerns, the submission does not pass the bar for acceptance.

**Justification For Why Not Higher Score:**

The reviewers were not in the support of acceptance.

**Justification For Why Not Lower Score:**

N/A

---

### Decision · Program_Chairs · 2024-01-16

Reject